# OpenReview forum: "Golden RPG: Semantic-Aware Noise for Regional Text-to-Image Generation"
_ICLR.cc/2026/Conference — ICLR 2026 Conference Withdrawn Submission_

### Official Review · Reviewer_NR3n · 2025-10-28

**Soundness:** 3
**Presentation:** 2
**Contribution:** 2
**Rating:** 2
**Confidence:** 5

**Summary:**

This paper presents Golden RPG, a framework that integrates semantic-aware Golden Noise into the RPG (Regional Planning) architecture to improve regional semantic consistency and compositional quality in text-to-image generation. The authors provide a systematic description of the architecture, Golden Noise generation mechanism, and regional prompt processing. Extensive qualitative and quantitative experiments, including metrics such as RSA (Regional Semantic Alignment), CRC (Cross-Region Coherence), and MOCQ (Multi-Object Composition Quality), as well as ablation studies, demonstrate the effectiveness of the proposed approach.

**Strengths:**

1. The overall method structure appears to be effective.

2. A reasonable system has been constructed to integrate Golden Noise with RPG.

3. The effectiveness of the method has been validated on a limited set of experiments.

**Weaknesses:**

1. Limited novelty: Although the authors constructed an effective system, it appears to be primarily a straightforward integration of Golden Noise and RPG.

2. Experiments and writing: Tables 1 and 2 contain essentially the same content, and the ablation results seem to be missing. In Section 5.2, the “Integration Necessity Justification” is already discussed, and then Section 5.4 repeats the same topic, leading to redundancy.

3. Insufficient experimental scope: For an engineering-focused paper, the authors could further explore whether the fusion of the two mechanisms yields a synergistic effect (i.e., 1+1>2), whether it meaningfully improves generation in complex scenarios, and whether there are any limitations.

**Questions:**

1. Could you clarify the contributions of your method?

2. Regarding Weaknesses 2, your presentation and writing require significant revision.

3. Could you provide some discussion addressing Weaknesses 3?

---

### Official Review · Reviewer_D8Ak · 2025-10-30

**Soundness:** 3
**Presentation:** 2
**Contribution:** 2
**Rating:** 4
**Confidence:** 4

**Summary:**

This paper proposed an enhanced RPG framework to integrate Golden Noise optimization to improve the regional semantic alignment, cross-region coherence, and multi-object composition.

**Strengths:**

1. The incorporation of RPG and Golden Noise is well motivated.

2. Quantitative improvements can be observed in experiments, including the proposed metrics, regional semantic alignment, cross-region coherence, and multi-object composition.

**Weaknesses:**

1. Lack of baselines: Only the Original RPG and Original Golden Noise are compared.  Missing comparison with other SOTA methods on standard benchmarks

2. Incremental contribution: the proposed method combines RPG and Golden Noise, where the techinical contributions require more clarification.

**Questions:**

1. Can the proposed method be a plug-and-play technique that can be used for other methods?

2. Fig. 1 is oversized with too many blank areas, which can be improved

---

### Official Review · Reviewer_L55Z · 2025-10-31

**Soundness:** 2
**Presentation:** 2
**Contribution:** 2
**Rating:** 4
**Confidence:** 3

**Summary:**

This paper proposes the Golden RPG framework, which deeply integrates semantically-aware Golden Noise optimization with the RPG paradigm. By replacing the original RPG’s uniform random noise initialization with Golden Noise–based initialization, the method enables a collaborative process between text-prompt-driven generation (strategic planning) and noise-prompt-driven generation (tactical execution). Experimental results demonstrate that the proposed approach surpasses baseline methods in terms of regional semantic alignment, inter-region coherence, and multi-object composition quality.

However, in my view, the paper essentially combines two existing, non-conflicting techniques—both of which are known to independently benefit text-to-image generation. Therefore, it is unsurprising that their integration leads to improvements over either method alone. As such, I believe the current contribution is incremental, and the work lacks substantial novelty.

**Strengths:**

The proposed method is conceptually reasonable and compatible with existing generative paradigms.

**Weaknesses:**

1. The approach primarily represents a straightforward combination of two existing methods, without introducing a fundamentally new algorithmic insight; the level of innovation appears limited.

2. There are inconsistencies in the claims. For instance, the Introduction states that the proposed method achieves a 24% accuracy improvement over GLIGEN, yet the experimental section does not provide any direct quantitative or qualitative comparison against GLIGEN. Instead, comparisons are limited to Original RPG, Original Golden Noise, and the proposed Golden RPG, leaving the “better than GLIGEN” claim unsupported.

3. Several figures in the paper appear ad hoc and lack clarity. For readers unfamiliar with the underlying techniques, it is difficult to comprehend the complete end-to-end pipeline.

**Questions:**

1. In the Introduction, you claim a 24% accuracy improvement over GLIGEN. However, I could not find corresponding experimental evidence in the results section. Could you clarify this discrepancy?

2. Since GLIGEN, several more advanced layout-to-image methods have been proposed. Why did you not include these methods as baselines in relevant layout-to-image benchmarks to more convincingly demonstrate the effectiveness of your approach?

---

### Official Review · Reviewer_9ASe · 2025-11-03

**Soundness:** 2
**Presentation:** 2
**Contribution:** 1
**Rating:** 0
**Confidence:** 5

**Summary:**

The paper combines two prior paradigms in diffusion-based text-to-image generation:

RPG (Recaptioning–Planning–Generating; Yang et al., 2024a) for regional decomposition, and

Golden Noise (Zhou et al., 2024) for semantic-aware noise optimization.

The proposed Golden RPG replaces uniform random noise with region-specific “golden noise” aligned to each sub-prompt, claiming improved regional semantic alignment and multi-object coherence. Experiments on Stable Diffusion XL report moderate quantitative gains (~24–36%) across three metrics.

**Strengths:**

Clear motivation: combining text-prompt planning with noise-prompt optimization.

Implementation appears straightforward and reproducible (plug-and-play with pretrained NPNet).

Results include both quantitative metrics and visual examples demonstrating visible improvements.

**Weaknesses:**

1. Limited novelty (integration of existing systems).

The method is not conceptually new. RPG already performs regional decomposition and regional diffusion (Yang et al., 2024), while Golden Noise (Zhou et al., 2024) already introduces semantic-aware noise priors.
Golden RPG merely applies Golden Noise per region inside RPG’s pipeline. This is an intuitive extension, not a new idea or algorithmic contribution.
Furthermore, region- or layout-conditioned diffusion is a well-developed line of work — ControlNet (Zhang et al., 2023), T2I-Adapter (Mou et al., 2023), and GLIGEN (Li et al., 2023) already achieve regional or grounding-based control.
The paper’s main step—injecting Golden Noise region-wise—constitutes compositional reuse, not innovation.

2. Conceptual framing without substance.

The framing (“strategic + tactical synergy,” “semantic–regional dual paradigm”) is rhetorical rather than technical.
There is no theoretical analysis explaining why region-wise Golden Noise should outperform existing layout-based control, nor how it differs mechanistically from simply conditioning noise via spatial embeddings.
The discussion repeatedly markets the framework as “paradigm bridging,” yet every module (noise learning, regional planning, prompt decomposition) already exists.

3. Weak experimental support.

Metrics: no human or standardized benchmark evaluation.

Baselines: Missing key modern controllable diffusion methods such as ControlNet, T2I-Adapter, or GLIGEN, which are more natural points of comparison.

Results: Improvements are small (e.g., CLIP score +0.05–0.1)

4. Lack of theoretical or empirical depth.

The paper does not attempt to characterize what “semantic-aware noise” encodes or how it interacts with the diffusion trajectory.
No examination of cross-region consistency, variance structure, or interpretability is given — leaving the method as an empirical trick, not a studied mechanism.

**Questions:**

N/A

---

### Note · Authors · 2026-01-27

**Comment:**

withdraw

**Withdrawal Confirmation:**

I have read and agree with the venue's withdrawal policy on behalf of myself and my co-authors.

---

### Meta-Review · Area_Chair_AgdR · 2025-12-17

**Summary:**

he reviewers recommend a Reject based on the following three core issues:

Limited Novelty: The work is viewed as a straightforward engineering integration of two existing paradigms (RPG and Golden Noise) without introducing fundamental algorithmic innovations or a unique theoretical contribution.

Weak Experimental Validation: The evaluation lacks comparisons against modern state-of-the-art baselines like ControlNet or GLIGEN, and it contains unsupported quantitative claims that are not backed by the provided data.

Insufficient Depth and Clarity: The paper relies on rhetorical framing rather than technical analysis to explain the "synergy" of the methods, and the presentation is marred by redundant text and poorly organized figures.

There is no rebuttal from the author so that this paper will be rejected

**Reviewer Concerns:**

No authors response so that this paper is rejected meaning that no concerns have been addressed

**Reviewer Scores:**

All reviewers are with reject.

---

### Decision · Program_Chairs · 2026-01-26

Reject